# Zoonotic *Giardia duodenalis* Genotypes and Other Gastrointestinal Parasites in a Badger Population Living in an Anthropized Area of Central Italy

**DOI:** 10.3390/pathogens11080906

**Published:** 2022-08-11

**Authors:** Michela Maestrini, Federica Berrilli, Alessia Di Rosso, Francesca Coppola, Isabel Guadano Procesi, Alessia Mariacher, Antonio Felicioli, Stefania Perrucci

**Affiliations:** 1Department of Veterinary Sciences, University of Pisa, Viale delle Piagge n. 2, 56124 Pisa, Italy; 2Department of Clinical Sciences and Translational Medicine, University of Rome “Tor Vergata”, 00133 Rome, Italy; 3PhD Program in Evolutionary Biology and Ecology, Department of Biology, University of Rome “Tor Vergata”, 00133 Rome, Italy; 4Istituto Zooprofilattico Sperimentale delle Regioni Lazio e Toscana, 58100 Grosseto, Italy

**Keywords:** Eurasian badger (*Meles meles*), anthropogenic settlement, gastrointestinal parasites, zoonotic *Giardia duodenalis* genotypes, central Italy

## Abstract

The Eurasian badger (*Meles meles*) is widespread in Italy and occupies different habitats. The occurrence and species of gastrointestinal parasites were evaluated in a free-ranging badger population living in a highly anthropic area in central Italy. A total of 43 fecal samples were examined using the flotation test, the Mini-FLOTAC and Baermann techniques, and a rapid immunoassay for the detection of *Giardia duodenalis* and *Cryptosporidium* spp. fecal antigens. Molecular investigations were also performed that aimed at identifying *Giardia* genotypes. Overall, 37/43 samples (86%) were found positive. Specifically, 48.8% (21 samples) were positive for *G.*
*duodenalis*, 23.2% (10/43) for *Cryptosporidium* spp., and 7% (3/43) for coccidian oocysts. *Strongyloides* sp. nematode larvae were detected in 3/43 samples (7%). Ascarid (1/43, 2.3%), capillariid (1/43, 2.3%), and strongyle-type eggs (76.7%, 33/43) were also identified. Among the 11 readable sequences of samples that were positive for *G. duodenalis* by end-point PCR (18/21), the zoonotic assemblage A sub-assemblage AII and mixed assemblage A and B were identified. This is the first report of zoonotic *G. duodenalis* genotypes in the Eurasian badger. Moreover, most of identified parasites have zoonotic potential and/or potential impact on the population health of wild badgers and other wild and domestic animals.

## 1. Introduction

The Eurasian badger (*Meles meles*) is an opportunistic omnivorous meso-carnivore of the Mustelidae family, mainly feeding on a wide variety of plants and animal foods (i.e., earthworms, large insects, small mammals, carrion, cereals, and tubers) [1,2,3]. Badgers are social, and in a high-density population this mustelid species lives in a mixed-sex clan composed of up to 12 individuals, while it adopts a more solitary lifestyle in a low-density population [3,4]. Badgers are semi-fossorial and mainly nocturnal mammals and use burrow systems (i.e., settlements) as main sites for overwintering, breeding, and sleeping during daylight hours [4,5,6]. They usually share the same settlement at different times with other semi-fossorial mammals, such as the crested porcupine (*Hystrix cristata*) and red fox (*Vulpes vulpes*), and occasionally can cohabit with the crested porcupine [7]. Except for the Italian islands, the Eurasian badger is widespread throughout all mainland Italy, where it can occupy different habitats, such as mountainous areas and the Mediterranean coastal zone but also suburban, cultivated, and anthropized areas (i.e., riverbanks, gardens, and city parks) [8,9,10,11]. Although the species is widespread in Italy, few data are available on the gastrointestinal parasites of badger populations in the country [12,13,14]. In Italy, the nematodes *Physaloptera sibirica*, *Uncinaria criniformis*, *Aonchotheca putorii*, and *Molineus patens* and the cestode species *Mesocestoides melesi* have been identified in badgers [12,13,14]. However, in other European countries, several gastrointestinal parasite species have been identified in badger populations [15,16,17,18,19,20,21], including parasite species that can also infect humans and/or other wild and domestic animal species, such as *Baylisascaris melis* described for the first time in badgers in Belgium [22], *Cryptosporidium* spp., and *Giardia duodenalis* [17,18]. Nevertheless, data on these badger parasite infections are completely lacking in Italy. In Italy, *G. duodenalis* infection has been detected in several wild animal species, such as the wolf (*Canis lupus italicus*), wild boar (*Sus scrofa*), Alpine chamois (*Rupicapra rupicapra rupicapra*), Apennine chamois (*Rupicapra pyrenaica ornata*), and crested porcupine (*H. cristata*) [23,24,25,26]. Moreover, potentially zoonotic assemblages and sub-assemblages have been identified in some of these animal species, such as *R. r. rupicapra*, *R. p. ornata*, and crested porcupines, living in both wild and anthropized areas of central Italy [23,25,26], where the Eurasian badger is also present. Therefore, this study was primarily designed to update data on gastrointestinal parasites infecting the Eurasian badger in Italy and to also evaluate the occurrence and genotypes of *G. duodenalis* in a Eurasian badger population living in an anthropized area of central Italy.

## 2. Results

Overall, 37 out of 43 (86%) samples were positive for gastrointestinal parasites at the parasitological analysis performed. Eight different parasite taxa were identified (Table 1). At the immunoassay, 48.8% (21/43) of the analyzed fecal samples were positive for *G. duodenalis* and 23.2% (10/43) for *Cryptosporidium* spp. Based on morphology, two types of coccidian oocysts (7%, 3/43, 15–55 oocysts per gram of feces (OPG)) were evidenced by the flotation and the Mini-FLOTAC techniques: the first type of oocyst was ovoidal, about 32 µm × 26 µm in size (Table 1, Figure 1A), which was detected in two samples, while in the third positive sample, oocysts were oval/elliptical and measured about 20 µm × 16 µm. At the flotation test and the Mini-FLOTAC technique, positivity for gastrointestinal strongyle-type eggs (76.7%, 33/43) (Table 1, Figure 1B,E) with a mean number of 84.8 eggs per gram of feces (EPG; range 10–800 EPG), capillariid eggs (2.3%, 1/43, 10 EPG) (Table 1, Figure 1D), and ascarid eggs (1/43, 2.3%, 10 EPG) was also found. Among the strongyle-type eggs, *Uncinaria criniformis* was identified in all samples (Table 1, Figure 1B). However, unidentified strongyle-type eggs were also detected in three badger fecal samples (3/43, 7%), which were concurrently positive for *U. criniformis* (Table 1, Figure 1E). Finally, *Strongyloides* spp. larvae were detected using the Baermann test in 3 out of 43 (7%) examined samples (Table 1, Figure 1C).

Multiple parasites were found in 14/37 positive samples (37.8%), most of which were coinfections between *Giardia* plus other parasites (11/43, 25.6%). More specifically, *Giardia* + strongyle-type eggs and *Giardia* + strongyle-type eggs + *Strongyloides* sp. larvae were detected in eight and two *Giardia*-positive samples, respectively. In the remaining *Giardia*-positive sample, gastrointestinal strongyle-type eggs + *Strongyloides* sp. + coccidia were identified.

Out of 21 samples that resulted *Giardia*-positive using the immunoassay, 18 were positive at end-point PCR and 11 readable sequences were obtained. In detail, at the *bg* locus, six samples were assigned to assemblage A sub-assemblage AII. Moreover, the presence of double peaks in diagnostic positions allowed us to identify a mixed assemblages A and B infection in one sample. As for the *tpi* locus, two samples were assigned to assemblage A sub-assemblage AII. Finally, one sample was successfully amplified for both genes and identified as assemblage A sub-assemblage AII at the *bg* locus and as mixed assemblages A and B infection at the *tpi* locus. Raw data (FASTA sequences) are reported in Appendix A.

## 3. Discussion

Gastrointestinal parasites may influence mustelid population size by reducing host fitness; lowering host body condition and fecundity, causing mortality; and changing host behavior or vulnerability to predation [27,28]. In a study performed in Switzerland on the causes of mortality and morbidity in free-ranging mustelids [16], gastrointestinal parasites were found to be associated with pathological changes of the digestive system in about 10% of the infected animals. The significance of the negative impact of gastrointestinal parasites on mustelid populations may be related to specific parasite species and/or infection intensity [28,29]. Nonetheless, in Italy, the Eurasian badger is not an endangered species, and parasites may also represent an important factor limiting the population growth of this wild animal. However, even if some gastrointestinal parasites are specific to a particular mustelid host and their circulation is therefore related to the dynamics of that host population [28,30], some mustelid gastrointestinal parasite species can infect multiple mustelids and other animal host species [22,31,32] including humans, thus having zoonotic significance [18,22,32]. In an anthropized environment, such as the area selected in this study, the risk of transmission of parasites from wild animals to humans and domestic animals and vice versa can be high [33], and this possibility strongly increases the relevance of the Eurasian badger population here examined in the epidemiology of potentially zoonotic parasites.

Among the protozoa, *G. duodenalis* infection was prevalent in the examined badger population, considering that about 50% of the examined samples (21/43) were positive for *G. duodenalis* according to the results of the rapid immunoassay. Moreover, the results of molecular analysis confirmed the positivity for this species in 18/21 samples. Only a previous study reported *G. duodenalis* identification in badgers, and this protozoan enteropathogen was suspected to be responsible for poor conditions and chronic diarrhea in rescued Eurasian badger cubs in the U.K. [19]. In free-ranging badgers, however, no positivity for *G. duodenalis* was found both in a single badger examined in Poland [34] and 70 badgers examined in Spain [18]. In the badger cub found positive in the U.K. [19], the results of molecular analysis revealed *G. duodenalis* assemblage E, while in the present study, genotyping of 11 PCR positive samples revealed potentially zoonotic assemblages (A and B) and sub-assemblages (A-II). In fact, molecular studies showed that *G. duodenalis* includes eight distinct genotypes, also called assemblages, identified with the alphabetic letters from A to H [35]. Assemblages A and B are more frequently found in humans, but they can also infect animals and are considered potentially zoonotic [36]. Assemblages A and B have been further divided into sub-assemblages, some of which are more common in humans and some in animals. Sub-assemblage A-II is considered human-specific, although it has also been detected in animals [36,37]. Assemblage B is genetically more polymorphic than assemblage A, and this makes assignment to specific genotypes more difficult [36,38]. Therefore, this study shows for the first time that Eurasian badgers can be infected with zoonotic genotypes of *G. duodenalis*. Consequently, this wild animal species may play a role in the environmental contamination with *G. duodenalis* zoonotic genotypes that can be responsible for infections in other wild and domestic animals and humans, thus extending the public health relevance of this mustelid species to *G. duodenalis*. The badger population examined in this study lives in a highly anthropized area in burrows located near gardens, vegetable gardens, etc. In developed countries, human infections caused by zoonotic *G. duodenalis* genotypes are mainly waterborne infections [39]. Therefore, although defecation of badgers takes place in latrines, some atmospheric events, among which are mainly the torrential rainfalls being observed in Italy in recent years [40,41], can easily cause the spread of cysts of these zoonotic *G. duodenalis* assemblages in the surrounding environment and, consequently, the easy contamination of vegetable gardens, gardens, and surface water used for human consumption [39]. Furthermore, a recent study performed in this area [7] reported that badgers cohabited or shared the same burrows with crested porcupines and foxes. More specifically, they cohabited with porcupines in the same burrow system in 43% of the cases, and interactions between adult badgers and porcupettes were also recorded [7]. Consequently, *G. duodenalis* cross-infections between crested porcupines, foxes, and badgers can be very likely in this area. Crested porcupines and foxes may be susceptible to infections with zoonotic assemblages of *G. duodenalis* [26,42] and, compared with badgers, have a habit of being closer to human settlements, as they frequently visit vegetable gardens, gardens, henhouses, etc., and they can easily contaminate these areas with zoonotic *G. duodenalis* genotypes, causing the passage of these zoonotic assemblages from the wild to the domestic environment. Moreover, in the examined area, crested porcupines (*H. cristata*) were also found to be infected by potentially zoonotic *G. duodenalis* genotypes [26].

In previous European studies in wild mustelids [18,43], *Cryptosporidium* infection was reported in American minks (*Neovison vison*) in Denmark and badgers in Switzerland and Spain [16,18], with a 3% prevalence found in the badgers examined in Spain [18]. In the present study, 23.2% of badger fecal samples were positive for *Cryptosporidium* spp. at the immunoassay results. Although it is likely that samples from the same badgers may have been collected and examined more than one time during the study period, results obtained in this study may be indicative of a high frequency of *Cryptosporidium* infection in the badger population here examined with respect to that reported in previous studies in other European areas. In Spain [18], *Cryptosporidium parvum* and *Cryptosporidium hominis* zoonotic species were identified after a molecular analysis was performed in positive badgers. Therefore, further molecular studies should be performed on *Cryptosporidium*-positive badger samples in this study to evaluate the zoonotic significance of this finding and the potential role of the badger population here examined in human *Cryptosporidium* infections.

In the case of coccidia, the Eurasian badger can be infected by the species *Eimeria melis* and *Isospora melis* [21,44]. *E. melis* infection occurs at higher intensity and prevalence in cubs than in adults, and it is considered a significant cause of mortality in cubs, although of less pathological significance in adults [21,44]. *I. melis* seems to have limited pathological significance both in adults and cubs, but coinfections between *E. melis* and *I. melis* are frequently observed in adult badgers, and a direct relationship between the intensity of the two species has been also evidenced [21,44]. Based on morphological features, coccidian oocysts here detected in the fecal samples of two coccidia-infected badgers were identified with *I. melis*, while the third positive badger was infected by *E. melis*. In Europe, both these coccidian species have been previously reported in badgers only in England [21,44].

Gastrointestinal nematodes previously reported in European badgers include the species *U. criniformis, Placoconus lotoris, Spirocerca lupi, Mastophorus muris, Aonchotheca putorii, Molineus patens, Physaloptera sibirica, Vigisospirura potekhina hugoti, Baylisascaris melis*, and *Strongyloides* sp. [13,14,15,17,20,45]. Among these nematode infections, positivity for gastrointestinal strongylid eggs were prevalent in the examined badger population, as 76.7% of badger fecal samples here examined were positive for these nematodes, although with a low mean EPG number (84.8 EPG). Based on egg morphology and size, *U. criniformis* was identified in all positive samples, confirming the data of previous reports about the frequent occurrence of this nematode species in the Eurasian badger [45,46,47]. *U. criniformis* is a blood-feeding nematode species belonging to the Ancylostomatidae family, which is primarily found in badgers and closely related to *Uncinaria stenocephala*, a pathogenic species of wild and domestic canids and felids [15,20,46]. However, three samples were concurrently positive for strongylid eggs that were morphologically different from those of *U. criniformis* (Figure 1B,E) and probably belonging to *M. patens*, a gastrointestinal strongyle species previously reported in Italian badgers [13,14].

Two examined badger fecal samples were positive for the presence of eggs of other nematode taxa. More specifically, a sample was positive for 10 EPG of ascarid eggs and a further sample for 10 EPG of capillariid eggs. Capillariid eggs were 59–63 µm in length and 22–26 µm in width and showed a striated shell with fine net-like ridges on the outer surface and two protruding plugs. Based on these features, the detected eggs were identified as *A. putorii*, a capillariid species that is mainly found in the stomach and less frequently in the small intestine of wild and domestic carnivore hosts, and that is considered a potential cause of gastritis in pet carnivores [31]. In mustelids, this capillariid species may have a negative impact by reducing body condition and probably affecting population sizes [28]. In previous studies, this species was reported in European badger populations in Spain, Poland, and northwestern Italy [14,45,47].

In the case of the ascarid-positive badger sample, *B. melis* is the only ascarid species known to infect European badgers, and it was first described in Belgium [22]. *Baylisascaris* is a nematode genus infecting the small intestine of carnivores, omnivores, and herbivores, but a wide range of other animal species may act as paratenic hosts, in which these nematodes may cause visceral, ocular, and neural *larva migrans* [22,32,48]. However, no confirmed cases of natural *larva migrans* caused by *B. melis* have been reported, although this species was found to cause visceral, ocular, and neural *larva migrans* in experimentally infected rodents [22]. Nonetheless, this species is included among the potential causes of baylisascariasis in humans, along with other *Baylisascaris* species [45]. Among these, *Baylisascaris procyonis* infecting raccoons (*Procyon lotor*) is considered the most important, as this species may cause severe neurologic disease in humans and numerous animal species [22]. *B. procyonis* has been reported in raccoons in the USA, Canada, and many European countries [30,49,50,51,52,53,54], including an area of central Italy very near to that of the badger population examined in this study [55]. Further molecular and epidemiological studies are needed to establish if the ascarid found in this single badger sample is effectively *B. melis* and to evaluate the frequency of occurrence and diffusion of *B. melis* infection in the examined badger population.

Although *Strongyloides* infections have been frequently reported in badgers, *Strongyloides* infecting European badger populations have never been identified at the species level [17,20]. In Europe, the species *Strongyloides procyonis*, *Strongyloides martis, Strongyloides mustelorum*, and *Strongyloides lutrae* have been reported in other mustelid species [30,56,57,58]. The pathogenicity of *Strongyloides* species infecting mustelids is still unknown, although it is assumed that respiratory distress can result from migration of *Strongyloides* spp. larvae through the lungs [59]. Interestingly, *S. procyonis* was demonstrated to cause experimental creeping eruption and a short-lived intestinal infection in inoculated human volunteers [33]. Moreover, the morphological parameters of *Strongyloides* sp. found in the Japanese badger (*Meles anakuma*) seemed to be comparable to those of *S. procyonis* and *S. martis*, but data from a phylogenetic study suggested that it was a species separate from *S. procyonis* [60].

## 4. Materials and Methods

### 4.1. Study Area and Sampling

Sampling was performed in a hilly area of 714 ha located in Crespina-Lorenzana and Lari-Casciana Terme (10.56815° N–43.56796° E) in the province of Pisa (Tuscany, central Italy) (Figure 2). The study area included 18 towns, with an average human density of 134.08 people/km^2^. The area was characterized by a highly anthropic fragmented agroecosystem characterized by small woody areas interspersed with agricultural and urban zones and interconnected by a dense network of ecological paths [61]. A wide variety of wild mammals lived in the study area, such as crested porcupines (*H. cristata*), wild boars (*Sus scrofa*), roe deer (*Capreolus capreolus*), pine martens (*Martes martes*), stone martens (*Martes foina*), skunks (*Mustela putorius*), badgers (*Meles meles*), hares (*Lepus europeus*), eastern cottontails (*Sylvilagus floridanus*), wild rabbits (*Oryctolagus cuniculus*), red foxes (*V. vulpes*), wolves (*Canis lupus*), and the introduced invasive coypu (*Myocastor coypus*) [61].

Badger fecal sampling was performed from September 2020 to April 2021 from latrines. Badger latrine detection was performed along 4 transects (2 SD 1.2 km), randomly chosen within the study area, for a total length of 8 km (Figure 2). Every transect was covered once per week, and from each detected latrine, two aliquots of each fresh fecal sample present were collected and pooled. From each latrine, only samples judged fresher were collected for parasitological analysis, avoiding collecting dry or moldy fecal samples. The freshness state of fecal samples was assigned based on external appearance and local weather conditions. Badger fecal samples were discriminated according to their deposition site (i.e., latrines), size, shape, and composition (i.e., amorphous, part of small invertebrates, and seeds) [62,63]. Collected fecal samples were stored at 4 °C and analyzed within 24 h.

### 4.2. Parasitological Analysis

Fecal samples were examined by using a commercial rapid immunoassay to search for *Giardia* and *Cryptosporidium* spp. fecal antigens (Rida Quick^®^
*Cryptosporidium*/*Giardia* Combi, R-Biopharm, Darmstadt, Germany). Moreover, for the detection and quantification of gastrointestinal nematode eggs and coccidian oocysts, all the fecal samples were examined by the flotation test and Mini-FLOTAC technique on 2 g of feces using saturated sodium chloride as the flotation solution (NaCl, specific gravity 1.2) [64]. Results are expressed as the arithmetic mean number of eggs/(oo)cysts per gram (EPG/OPG) of feces [64]. The Baermann test [65] was used for the detection of *Strongyloides* sp. larvae. Magnifications of 100× and 400× were used to identify nematode eggs/larvae and protozoan oocysts, which were measured under an optical microscope by using a micrometric eyepiece. Microscopic parasite identification was based on morphological and metric data on badger gastrointestinal nematodes and protozoa reported in previous studies [15,20,21,28,30,31,44,45,46,47].

### 4.3. Molecular Analysis

In the case of *Giardia*-positive samples using the immunoassay, molecular analysis was performed to identify *Giardia* species and genotypes. For DNA extraction, samples were processed using a commercial kit (QIAamp DNA Stool Mini Kit, QIAGEN, Valencia, CA, USA). PCR protocols were applied to amplify fragments of the small subunit ribosomal RNA (*SSU rRNA*, 130 bp) [66], ß-*giardin* (*bg*, 384 bp) [67], and triose phosphate isomerase (*tpi*, 530 bp) genes [68]. Positive amplicons were purified using the mi-PCR Purification Kit, Metabion International AG (Planegg/Steinkirchen, Munich, Germany). Amplification products were sent to an external laboratory for sequencing (Bio-Fab Research, Rome, Italy). Forward and reverse sequences were manually checked using Finch TV 1.4 software (Geospiza, Inc., Seattle, WA, USA). The obtained consensus sequences were then compared to those available in the GenBank database using the Standard Nucleotide BLAST search.

## 5. Conclusions

The data obtained in this study broaden our knowledge of gastrointestinal parasites of the Eurasian badger in Europe, and this is the first report of zoonotic *G. duodenalis* genotypes in this wild animal species. Moreover, *G. duodenalis*, *Cryptosporidium* spp., coccidia, *Strongyloides* sp., and ascarids were recorded for the first time in the Eurasian badger in Italy. Several of the parasites identified in this study have zoonotic potential and/or potential impact on the population health of wild badgers and other wild and domestic animals, thus highlighting the public and animal health relevance of this mustelid species.

## Figures and Tables

**Figure 1 pathogens-11-00906-f001:**
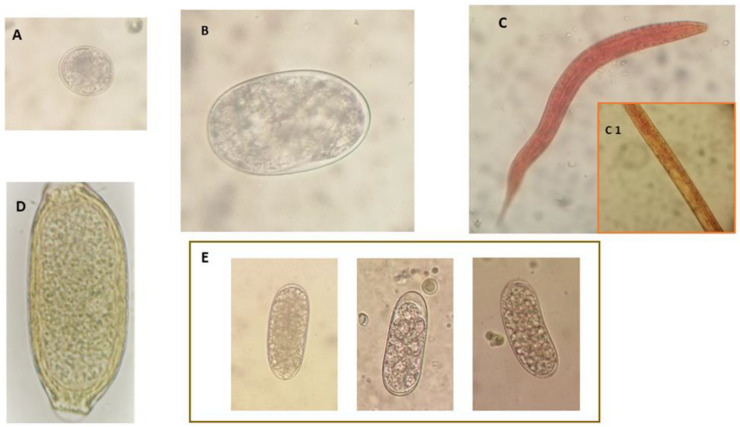
Digestive tract parasites identified in fecal samples from a Eurasian badger population in Tuscany, central Italy. (**A**) *Isospora melis* unsporulated oocyst; (**B**) *Uncinaria criniformis* egg; (**C**) *Strongyloides* sp. first stage larva particularly of the rhabditoid esophagus (**C1**); (**D**) *Aonchoteca putorii* egg; (**E**) unidentified strongyle-type nematode eggs.

**Figure 2 pathogens-11-00906-f002:**
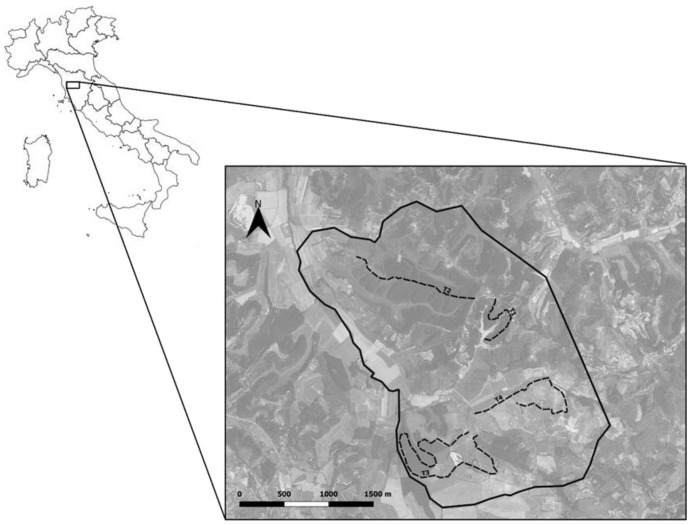
Map of Italy: the inset detail of the study area in Crespina-Lorenzana and Lari-Casciana Terme, Tuscany, central Italy (black line border), and the four transects (dashed black lines) in which badger fecal samples were collected from latrines.

**Table 1 pathogens-11-00906-t001:** Gastrointestinal parasites detected in 43 fecal samples of a European badger (*Meles meles*) population living in an anthropized area in central Italy.

Parasites	No. Positive Samples (%)	EPG/OPG * (Range)
**Helminths**		
*Uncinaria criniformis* eggs ^a^	33/43 (76.7%)	84.8 (10–800)
Unidentified gastrointestinal Strongyle-type eggs ^a^	3/43 (7%)	10 EPG
Capillariid eggs ^a^	1/43 (2.3%)	10 EPG
Ascarid eggs ^a^	1/43 (2.3%)	10 EPG
*Strongyloides* sp. ^b^	3/43 (7%)	-
**Protozoa**		
*Giardia duodenalis* ^c^	21/43 (48.8%)	-
*Cryptosporidium* spp. ^c^	10/43 (23.2%)	-
Coccidian oocysts ^a^	3/43 (7%)	10–55 OPG

***** EPG/OPG: eggs/oocysts per gram of feces; **^a^** parasites detected by using flotation test and Mini-Flotac technique; **^b^** parasites detected by using Baermann test; **^c^** parasites detected by using a commercial immunoassay to search for *Giardia* and *Cryptosporidium* spp. fecal antigens.

## Data Availability

Not applicable.

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
