# Peer review of "Zoonotic Giardia duodenalis Genotypes and Other Gastrointestinal Parasites in a Badger Population Living in an Anthropized Area of Central Italy"

_pathogens, 2022, doi:10.3390/pathogens11080906_

Round 1
Reviewer 1 Report
Comments pathogens-1850720
The manuscript pathogens-1850720 was reviewed. The MS is well written and gives an overview of the endo parasite presence of the badger in a region of Tuscany, Italy. Sampling of wildlife is always a challenge, but the authors did a good job by sampling a fair amount of samples in a defined area. The data are an addition to the knowledge of parasites in badgers in Italy. I have only some minor comments.
11) The assemblages A and B of Giardia are indeed zoonotic and the authors discuss this thoroughly. However, is this really a big problem? There are not too many badgers and they live mostly not very near human populations. What is the chance of obtaining an Giardia infection from badgers, compared to obtaining it from other humans or many pet animals? Can the authors discuss this point.
22)The authors indicate in many places in the MS the negative effect of the parasites. That can be true for an individual badger, but isn’t it so that the parasites, like predators also have a function on population level? Can the authors comment on this.
33) Some gene fragments of Giardia were sequenced, but I could not see that they have been submitted to any repository. Submission of the sequences to Genbank would be helpful for future research. In addition, I support sharing as much of the data as possible, so may be you can consider to add all the raw data in a spread sheet as supplementary data.
44) Line 72 stated that there are 8 different parasite taxa in table 1, but I counted only 7.
Author Response
Review 1 Report Form
Comments pathogens-1850720
The manuscript pathogens-1850720 was reviewed. The MS is well written and gives an overview of the endo parasite presence of the badger in a region of Tuscany, Italy. Sampling of wildlife is always a challenge, but the authors did a good job by sampling a fair amount of samples in a defined area. The data are an addition to the knowledge of parasites in badgers in Italy. I have only some minor comments.
Response: The authors thank Reviewer 1 for these comments.
11) The assemblages A and B of Giardia are indeed zoonotic and the authors discuss this thoroughly. However, is this really a big problem? There are not too many badgers and they live mostly not very near human populations. What is the chance of obtaining an Giardia infection from badgers, compared to obtaining it from other humans or many pet animals? Can the authors discuss this point.
Response: As specified in the manuscript, the badger population examined in this study live in a highly anthropized area in burrows located near gardens, vegetable gardens etc., etc. In developed countries, human infections caused by zoonotic Giardia duodenalis assemblages (together with those caused by the zoonotic species of Cryptosporidium) are mainly waterborne infections (see Ryan UM, Feng Y, Fayer R, Xiao L. Taxonomy and molecular epidemiology of Cryptosporidium and Giardia - a 50 year perspective (1971-2021). Int J Parasitol. 2021 Dec;51(13-14):1099-1119. doi: 10.1016/j.ijpara.2021.08.007). Therefore, although the defecation of badgers takes place in latrines, some atmospheric events, among which mainly the torrential rainfall that are being observed in Italy in recent years (see https://www.repubblica.it/argomenti/maltempo; https://www.lanazione.it/cronaca/maltempo-toscana-1.6848480), can easily cause the spread of cysts of these zoonotic G. duodenalis assemblages in the surrounding environment and, consequently, the easy contamination of vegetable gardens, gardens and of surface water used for human consumption (see http://www.arpat.toscana.it/datiemappe/dati/qualita-delle-acque-superficiali-destinate-alla-produzione-di-acque-potabili; see Ryan UM, Feng Y, Fayer R, Xiao L. Taxonomy and molecular epidemiology of Cryptosporidium and Giardia - a 50 year perspective (1971-2021). Int J Parasitol. 2021 Dec;51(13-14):1099-1119. doi: 10.1016/j.ijpara.2021.08.007 ). Furthermore, as already specified in the manuscript, it has been observed that the badgers of the examined area often cohabit or share the same burrows with crested porcupines and foxes (see Coppola, F.; Dari, C.; Vecchio, G.; Scarselli, D.; Felicioli, A. Cohabitation of settlements among crested porcupine (Hystrix cristata), red fox (Vulpes vulpes) and European badger (Meles meles). Curr. Sci. 2020, 119, 817-822.). These latter animal species may be susceptible to infections with the same zoonotic assemblages of G. duodenalis (Coppola, F.; Maestrini, M.; Berrilli, F.; Procesi, I.G.; Felicioli, A.; Perrucci, S. First report of Giardia duodenalis infection in the crested porcupine (Hystrix cristata L., 1758). Int J Parasitol Parasites Wildl. 2020, 11, 108-113. doi: 10.1016/j.ijppaw.2020.01.006; Hamnes IS, Gjerde BK, Forberg T, Robertson LJ. Occurrence of Giardia and Cryptosporidium in Norwegian red foxes (Vulpes vulpes). Vet Parasitol. 2007 Feb 28;143(3-4):347-53. doi: 10.1016/j.vetpar.2006.08.032.) and, compared to the badgers, have the habit to get closer to human settlements, as they frequent visit vegetable gardens, gardens, henhouses etc., etc., that they can easily contaminate with these zoonotic G. duodenalis genotypes, causing in this way the passage of these zoonotic assemblages from the wild to the domestic environment. This was added to the revised manuscript.
22)The authors indicate in many places in the MS the negative effect of the parasites. That can be true for an individual badger, but isn’t it so that the parasites, like predators also have a function on population level? Can the authors comment on this.
Response: The authors completely agree with this comment of the reviewer 1. The role of parasites as an important factor limiting wild animal population growth is in fact very well known. Moreover, the Eurasian badger is not an endangered species in Italy. The discussion of the revised manuscript was changed accordingly.
33) Some gene fragments of Giardia were sequenced, but I could not see that they have been submitted to any repository. Submission of the sequences to Genbank would be helpful for future research. In addition, I support sharing as much of the data as possible, so may be you can consider to add all the raw data in a spread sheet as supplementary data.
Response: We fully agree with the reviewer and will submit all sequences obtained in this study to GenBank as soon as possible as part of a larger ecological study on Tuscan wild animals. Also, to share our findings with researchers, we are very pleased to add all raw data as Supplements.
44) Line 72 stated that there are 8 different parasite taxa in table 1, but I counted only 7.
Response: This is right. The authors thank Reviewer 1 a lot for this comment. In the revised manuscript, the missing nematode has been added to Table 1.
Reviewer 2 Report
Review of Manuscript ID: pathogens-1850720
This article reports new information on parasites of an important semi-fossorial and mainly nocturnal mammals Eurasian badger (Meles meles) living in the central Italy. This is the first original discovery confirming the occurrence of zoonotic Giardia duodenalis genotypes in this wild animal species what means that the spectrum of giardiasis causative agents in humans is wider than originally thought. The research is based exclusively on fecal samples, which somewhat limits the ability to ascertain true prevalence and intensity of infections. This article is primarily of territorial and international importance, where the area of disease emergence among badgers represent important and growing field. Thus, this paper does have some broader implications, and is appropriate for publication in Pathogens.
The paper is generally well organized. The methods are adequate and the conclusions are well based on the results obtained. The pertinent literature is adequately reviewed. The photographic images are of good quality and of some value to the manuscript. The manuscript does not have many errors in the English language.
I have only one comment: How were the gastrointestinal nematode eggs and coccidian oocysts identified from the fecal samples?
I am finding the article as acceptable to the printing after minor revision.
Author Response
Review 2 Report Form
Review of Manuscript ID: pathogens-1850720
This article reports new information on parasites of an important semi-fossorial and mainly nocturnal mammals Eurasian badger (Meles meles) living in the central Italy. This is the first original discovery confirming the occurrence of zoonotic Giardia duodenalis genotypes in this wild animal species what means that the spectrum of giardiasis causative agents in humans is wider than originally thought. The research is based exclusively on fecal samples, which somewhat limits the ability to ascertain true prevalence and intensity of infections. This article is primarily of territorial and international importance, where the area of disease emergence among badgers represents important and growing field. Thus, this paper does have some broader implications, and is appropriate for publication in Pathogens.
The paper is generally well organized. The methods are adequate, and the conclusions are well based on the results obtained. The pertinent literature is adequately reviewed. The photographic images are of good quality and of some value to the manuscript. The manuscript does not have many errors in the English language.
Response: The authors thank a lot Reviewer 2 for these comments.
I have only one comment: How were the gastrointestinal nematode eggs and coccidian oocysts identified from the fecal samples?
Response: they were identified based on morphological and metrical features of badger gastrointestinal nematodes and protozoa reported in the previous literature. To be clearer, the same approach and fecal parasitological analysis that are used regularly to diagnose infections caused by gastrointestinal nematodes and protozoa in alive domestic and wild animals were used. However, in the revised manuscript the references used for parasite identification, which were reported only in the discussion of the original manuscript, have been added also in Materials and Methods.
I am finding the article as acceptable to the printing after minor revision.